# Electric and Magnetic Fields Effects in Vertically Coupled GaAs/Al$_x$Ga$_{1−x}$As Conical Quantum Dots



**Ana María López Aristizábal** *[ID], **Fernanda Mora Rey** *[ID], **Álvaro Luis Morales** [ID], **Juan A. Vinasco** [ID]
and **Carlos Alberto Duque** [ID]

Grupo de Materia Condensada-UdeA, Instituto de Física, Facultad de Ciencias Exactas y Naturales,
Universidad de Antioquia UdeA, Calle 70 No. 52-21, Medellín 050010, Colombia;
alvaro.morales@udea.edu.co (Á.L.M.); juan.vinascos@udea.edu.co (J.A.V.); carlos.duque1@udea.edu.co (C.A.D.)
* Correspondence: ana.lopeza1@udea.edu.co (A.M.L.A.); fernanda.morar@udea.edu.co (F.M.R.)

**Abstract:** Vertically coupled quantum dots have emerged as promising structures for various applications such as single photon sources, entangled quantum pairs, quantum computation, and quantum cryptography. We start with a structure composed of two vertically coupled GaAs conical quantum dots surrounded by Al$_x$Ga$_{1−x}$, and the effects of the applied electric and magnetic fields on the energies are evaluated using the finite element method. In addition, the effects are evaluated by including the presence of a shallow-donor impurity. The electron binding energy behavior is analyzed, and the effects on the photoionization cross-section are studied. Calculations are carried out in the effective mass and parabolic conduction band approximations. Our results show a notable dependence on the electric and magnetic fields applied to the photoionization cross-section. In general, it has been observed that both the electric and magnetic fields are useful parameters for inducing blueshifts of the resonant photoionization cross-section structure, which is accompanied by a drop in its magnitude.

**Keywords:** conical quantum dots; electric field; magnetic field; shallow-donor impurity; binding energy; photoionization





## 1. Introduction

The confinement of particles in regions with sizes on the order of nanometers has enabled the emergence of nanotechnology. Depending on whether the confinement is in one, two, or three dimensions, they are called quantum wells (QWs), quantum wires (QWWs), and quantum dots (QDs), respectively. The advantage of these structures lies in the ease of controlling the spectrum through size, shape, and inclusion of impurities or external effects such as electric, magnetic, and intense non-resonant laser fields. In the work [1], the authors varied the width of doped delta wells and included an electric field to study the effects on transport and quantum lifetimes of electrons. Priyanka et al. [2] studied the effects of hydrogenic impurity in quantum wires on nonlinear optical properties such as second- and third-harmonic generation. Moreover, they also included the presence of electric and magnetic fields and the contribution of Rashba spin–orbit interaction. In [3], Hayrapetyan performed a correction study on the biexciton spectrum of ellipsoidal quantum dots using the so-called Darwin term that could be treated as a perturbation.

A structure formed by coupled quantum dot-ring was modeled, and optical properties with changes in pressure, temperature, and in the presence of electric and magnetic fields were simulated in [4]. A fundamental part of their article was the modeling of a quantum ring-dot system taken from an experimental work [5], whose synthesis was performed using droplet epitaxy, and profile images were built from the measurements generated via Atomic Force Microscopy (AFM). In [6], a coupled quantum dot-ring system was simulated under combined electric and magnetic field effects, and exciton contribution studies were

also performed. Effects of hydrostatic pressure and temperature on a conical quantum dot with a spherical cap (ice cream type) are analyzed in [7]. By using CdSe/ZnS quantum dots, Dissanayake et al. [8] verified through an experimental development that photosynthesis in the microalga Chlorella Vulgaris is sensitive to the concentration of quantum dots. A review of 88 references carried out in [9], is dedicated to quantum dots as structures to facilitate early detection of cancer and used as mediators for drug delivery. Haleem et al. wrote a review paper on applications found in Scopus, Google Scholar, ResearchGate on medical applications of nanotechnology and mentioned the prospects of this field [10].

From three decades ago to the present, the study of QDs has increased as evidenced in several geometries and semiconductor materials studies [11–14]. Fomin et al. [15] developed a theory for excitonic interaction and compared it with experimental data of photoluminescence in CdSe quantum dots. In [16], the third-harmonic generation for cylindrical quantum dots in the presence of an electric field was studied analytically. A well-known electron and binding energy result for an off-center donor in a spherical quantum dot was obtained in [17]. There, the variational method is used to calculate the spectrum, and the authors conclude that the Coulombian center can drastically change the confinement of an electron in large-radius spheres. For conical structures, it has been shown that the electric field allows tuning the evolution of the structure from a quantum dot to a quantum ring [18]. For GaAs cone-shaped QDs under externally applied static electric field influence, the results of the energies of the electron states and the photoionization cross-section are reported as geometric parameters functions of the cone-shaped structures, as well as the intensity of the electric field [19].

There are a variety of studies for different QD geometries. Spherical QDs with parabolic confinement with an external electric field and in the presence of an impurity were presented in a previous work [20]. In triangular geometries, theoretical results show that in bilayer triangular graphene QDs with zigzag edges, the magnetism can be controlled by an external vertical electric field [21]. On the other hand, by using low-temperature microphotoluminescence spectroscopy, the effect of a lateral electric field on QD excitons trapped by single-layer width fluctuations in a narrow quantum well was investigated [22].

Various methods to model and numerically solve differential equations are used in these kinds of systems. Using the stabilization method, the system's eigenvalues are calculated with the effective mass approximation through the Raleigh–Ritz variational method [23]. Using the adiabatic approximation, strongly elongated and flattened conical QDs were investigated under external electric field effects [24]. The effective mass approximation in a two-band parabolic model was used to study hole and electron states in two truncated conical QDs considering the effects of the geometrical parameters [11]. Other studies use the finite element method and Arnoldi iterations to study low-dimensional heterostructures [25].

The photoionization cross-section (PCS) has previously been studied in QDs. The results of prior studies indicate that the PCS is influenced by the quantum size and impurity position. Graphically, the PCS is similar to the Gauss function curve, and the amplitude grows with the dot radius [26–28].

Vertically coupled quantum dots (VCQDs) have long been studied owing to their potential in the design of many devices. This is due to the easiness of changing the geometry of the individual dots and changing the coupling between them, which leads to applications including single photon sources, entangled quantum pairs, qubits and gates in quantum computation, quantum cryptography, solar cells, lasers, LEDS, biomedical imaging, and drug delivery systems. Also, VCQDs are studied for transport properties which leads to the applications mentioned above. Sargsian et al. [29] studied cylindrical VCQDs built from InAs in a GaAs matrix under the effect of an intense laser field and calculated the absorption coefficient, the refractive index changes, and second- and third-harmonic generation at different temperatures and laser field parameters. Makhlouf et al. [30] addressed the problem of calculating the nonlinear optical rectification for two vertically coupled layers of $In_xGa_{1-x}As$/GaAs, including lateral interaction with neighbor QDs. Their findings

show that indium segregation into the wetting layer affects the optical response significantly. In their work, the vertical coupling thicknesses also considerably affect the nonlinear optical rectification. In Ref. [31], Mei et al. measured the VCQD optical modal gain of CdZnTe/ZnTe, where the lower QD is smaller; the results showed that the peak gain of the coupled QDs, separated 6 nm, was found comparable to the large separation QDs, 18 nm. Tongbram et al. [32] researched on the multi-stacked vertically coupled InAs quantum dots capped by a combinational capping layer of InAlGaAs and GaAs layer. They found that the central part of the QD structure was stable in size, shape, composition, and density, which improves carrier confinement in the QDs. They propose that their structure can be employed to fabricate a single-photon source operating in the 1.3 μm telecom O-band.

This article focuses on a theoretical study of a vertically coupled "double conical quantum dot" (DCQD) of $Al_xGa_{1-x}As$ at a concentration of $x = 0.3$ under shallow donor impurity presence and electric and magnetic fields. The shape of this structure is a double conical QD composed of a lower cone and an upper cone, as illustrated in Figure 1. The structure has azimuthal symmetry, so if we rotate it about the $z$-axis, we obtain the three-dimensional perspective, also shown in Figure 1. The system of two vertically coupled conical quantum dots has been reported experimentally by Heyn and coworkers [33,34]. Using the local droplet etching (LDE) technique, where strain-free and widely adjustable GaAs quantum-dot molecules (QDMs) can be synthesized, they studied the excited-state indirect excitons in GaAs quantum dot molecules. Regarding the impurities located along the axial axis considered in this study, it is important to clarify that this is one of the most particular cases of the problem to be implemented. Although it is possible to establish an approximate region where impurities can be located within the structure, intentional doping is still a technique in development. A more extensive theoretical study should consider random doping with impurities within the structure, including acceptor impurities. The article is organized as follows: Section 2 presents the theoretical framework, Section 3 discusses the results, and Section 4 summarizes the study's main conclusions.

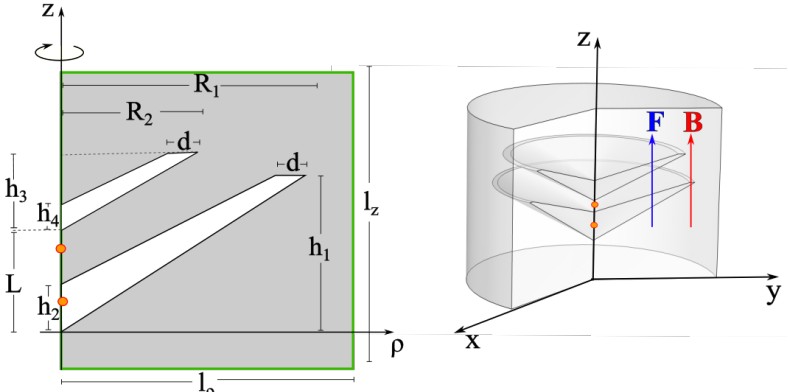

**Figure 1.** Vertically coupled double conical $GaAs/Al_{0.3}Ga_{0.7}As$ quantum dot with permanent dimensions: $R_1 = 40$ nm, $R_2 = 30$ nm, $d = 3$ nm, $L = 7$ nm, $h_1 = 20$ nm, $h_2 = 5$ nm, $h_3 = 15$ nm, $h_4 = 3$ nm and with boundary limits $l_r = 50$ nm of wide by $l_z = 40$ nm of high. On the right-hand side figure is the rotated two-dimensional structure. The orange dots denote the two impurity positions considered in this work: $z_i = 2.5$ nm and $z_i = 6.0$ nm. In left-hand side figure, the Dirichlet boundary conditions are considered along the green lines, which correspond to the cylindrical and two circular surfaces of the cylinder (see the right-hand side figure).

## 2. Theoretical Framework

In this work, we investigate the effects of axially applied electric and magnetic fields on the electronic structure of a confined shallow donor impurity in vertically coupled conical quantum dots. Figure 1 left shows the worked geometry corresponding to two vertically coupled conical quantum dots, both GaAs and surrounded by $Al_xGa_{1-x}As$ (in this work, all the reported calculations are for $x = 0.3$). Several impurity positions will be

considered along the $z$-axis to preserve the axial symmetry of the system. The coordinates of the origin are located at the lower vertex of the lower conical quantum dot ($z = 0$). On the other hand, Figure 1 right shows the $\phi = 0$ projection of the heterostructure, where the dimensions of the two quantum dots are indicated together with their spatial separation. In addition, the structure was subjected to the supply of static electric and magnetic fields, both in the axial direction and positive in the $z$-direction. The radius ($l_r$) and height ($l_z$) of the cylindrical region depicted in Figure 1 left have been chosen large enough to guarantee the convergence of at least the lowest ten energy levels. The Dirichlet boundary conditions are imposed on the surface of the cylindrical region in Figure 1 left.

In the effective mass and parabolic bands approximation, the one-band Hamiltonian for a confined electron in the double quantum dot heterostructure described above can be written as [11]

$$\hat{H} = \left[ -i\,\hbar\,\vec{\nabla} + e\,\vec{A}(\vec{r}) \right] \left[ \frac{1}{2\,m^*(\vec{r})} \right] \cdot \left[ -i\,\hbar\,\vec{\nabla} + e\,\vec{A}(\vec{r}) \right] + e\,\vec{F} \cdot \vec{r} - \frac{\kappa\,e^2}{4\,\pi\,\varepsilon\,\varepsilon_0\,|\vec{r} - \vec{r}_i|} + V(\vec{r}),$$ (1)

where $m^*(\vec{r})$ is the position dependent electron effective mass, $\vec{F} = F\,\hat{u}_z$ is the $z$-directed applied electric field, $\varepsilon$ is the GaAs static dielectric constant, $\vec{r}_i = z_i\,\hat{u}_z$ is the impurity position, $\kappa$ is a parameter which controls the presence ($\kappa = 1$) or absence ($\kappa = 0$) of the donor impurity, $e$ is the absolute value of the electron charge, $\vec{B} = \vec{\nabla} \times \vec{A}$ is the vector potential associated to $z$-directed applied magnetic field, $\vec{B} = B\,\hat{u}_z$, and $V(\vec{r})$ is the confinement potential associated to the heterostructure. Additionally, $\vec{\nabla} \cdot \vec{A} = 0$.

Considering that the system has axial symmetry, the corresponding wave function associated with the Hamiltonian in Equation (1) can be written in cylindrical coordinates as $\Psi(\vec{r}) = \Psi(\rho, z, \varphi) = \psi(\rho, z)\exp(i\,l\,\varphi)$, where $l$ is an integer number. By using the Hamiltonian in Equation (1) and the previously described wave function together with the Coulomb gauge, we arrive at the following two-dimensional Schrödinger equation in the cylindrical coordinates system [11]

$$\left\{ -\vec{\nabla}_{\rho,z} \left[ \frac{\hbar^2}{2\,m^*(\rho,z)} \right] \cdot \vec{\nabla}_{\rho,z} + \frac{\hbar^2\,l^2}{2\,m^*(\rho,z)\,\rho^2} + \frac{e\,\hbar\,B\,l}{2\,m^*(\rho,z)} + \frac{e^2\,B^2\,\rho^2}{8\,m^*(\rho,z)} \right.$$
$$\left. + e\,F\,z - \frac{\kappa\,e^2}{4\,\pi\varepsilon\,\varepsilon_0\,\sqrt{\rho^2 + (z - z_i)^2}} + V(\rho, z) \right\} \psi(\rho, z) = E\,\psi(\rho, z),$$ (2)

where $\vec{\nabla}_{\rho,z}$ is the $\rho$- and $z$-dependent two-dimensional gradient operator, with $\rho = \sqrt{x^2 + y^2}$.

Once Equation (2) has been solved to obtain the energy spectrum and the corresponding wave functions in the absence and presence of the donor impurity, this is for a particular configuration of the dimensions of the structure and the applied external fields. We proceed to calculate the binding energy for the ground state. In this study, the binding energy $E_b$ is defined as the difference between the first level on the electron without Coulomb interaction ($E_1^0$) and the first level on the electron with Coulomb interaction ($E_1^1$), i.e.,

$$E_b = E_1^0 - E_1^1.$$ (3)

Let us introduce the theoretical part of the PCS describing the transitions from the donor impurity ground state ($|\Psi_1^1\rangle$, with energy $E_1^1$) to the final state of the confined electron without the donor impurity effects ($|\Psi_1^0\rangle$, with energy $E_1^0$). In the dipole approximation, it is given by the expression [19,26,27].

$$\sigma(\hbar\,\omega) = \frac{4\,\pi^2\,\hbar\,\omega\,\alpha_{FS}}{n_r} \left( \frac{F_{eff}}{F_0} \right)^2 \left( \frac{m^*}{m_0} \right)^2 |\langle \Psi_1^1 | \vec{\xi} \cdot \vec{r} | \Psi_1^0 \rangle|^2 \delta(E_1^0 - E_1^1 - \hbar\,\omega),$$ (4)

where $\alpha_{FS} = \frac{e^2}{\hbar\,c}$ is the fine structure constant, $n_r$ is the refractive index of semiconductors, and $F_{eff}$ is the effective electric field on the impurity. Additionally, $F_0$ is the average field,

$m_0$ is the free electron mass, $\hbar\omega$ is the incident photon energy, and $\langle\Psi_1^1|\vec{\xi}\cdot\vec{r}|\Psi_1^0\rangle$ is the matrix element between the initial and final states of the impurity dipole moment, where $\vec{\xi}$ is the light wave polarization vector. In this study, we choose $\vec{\xi}=\hat{z}$, with this in mind $\vec{\xi}\cdot\vec{r}=z$. We note that in Equation (4) is used the effective and average fields of the light wave and not the externally applied constant electric field.

In addition, in this condition, the PCS may then be simplified by approximating the $\delta$–function by a Lorentzian one [19,26,27]

$$\sigma(\hbar\omega)=\sigma_0\frac{\Gamma\,I_{10}}{(E_b-\hbar\omega)^2+\Gamma^2}\,,\tag{5}$$

where the $\Gamma$-parameter is the hydrogenic impurity linewidth, $\sigma_0=\frac{4\pi\hbar\omega\,\alpha_{FS}}{n_r}\left(\frac{F_{eff}}{F_0}\right)^2\left(\frac{m^*}{m_0}\right)^2$, and $I_{10}$ represents the optical integral given by:

$$I_{10}=\left|2\,\pi\int_\Omega\left[\Psi_1^1(\rho,z)\right]^*z\,\Psi_1^0(\rho,z)\,\rho\,d\rho\,dz\right|^2,\tag{6}$$

where $\Omega$ extends over all space.

## 3. Results and Discussion

The parameters used in this work are: $x=0.3$, $m^*(GaAs)=0.067\,m_0$, $m^*(Al_xGa_{1-x}As)=0.081\,m_0$, $V(GaAs)=0$, $V(Al_xGa_{1-x}As)=227$ meV, $\varepsilon=12.65$, $\Gamma=3.0$ meV, $n_r=\sqrt{\varepsilon}$, and $F_{eff}/F_0=1$ [11,19,26,27]. Here, $m_0$ is the free electron mass. Calculations are at $T=4$ K. For finite temperature values, the different filling of the initial and final states can affect the photoionization cross-section.

Figures 2–8 display the results for the energy eigenvalues with applied external electric and magnetic fields and the ground state probability density with and without the presence of a donor impurity (at $z_i=2.5$ nm). In Figures 3 and 4, the blue color corresponds to zero magnitude, whereas the red one represents the maximum magnitude.

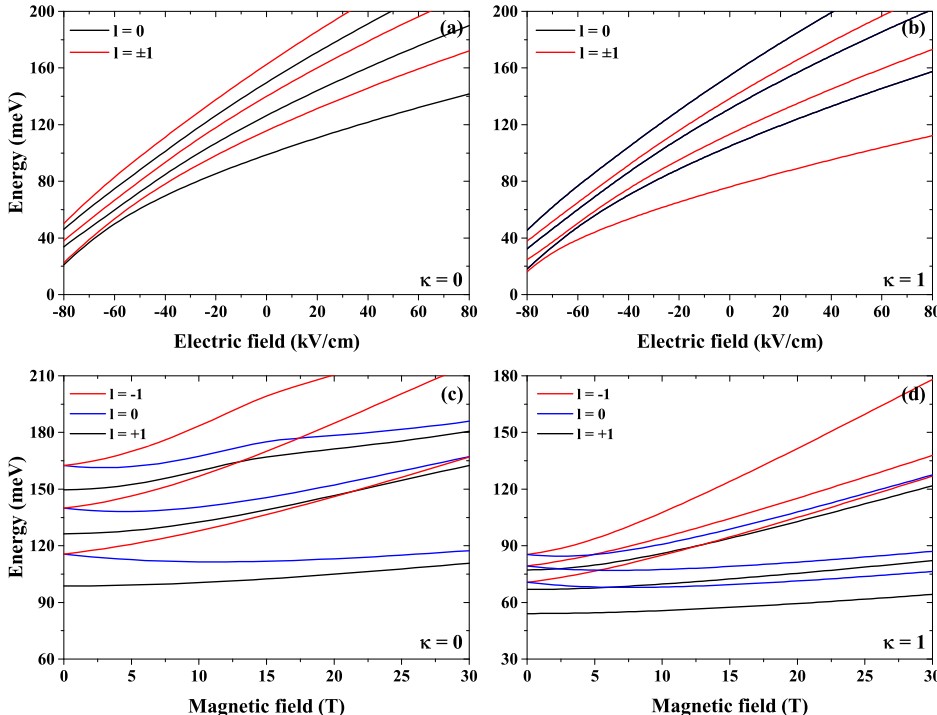

**Figure 2.** The energy of the lowest confined electron states in GaAS DCQD under the electric field effect with/without donor impurity (**a**,**b**). In panels (**c**,**d**), the results are a function of the applied magnetic field.

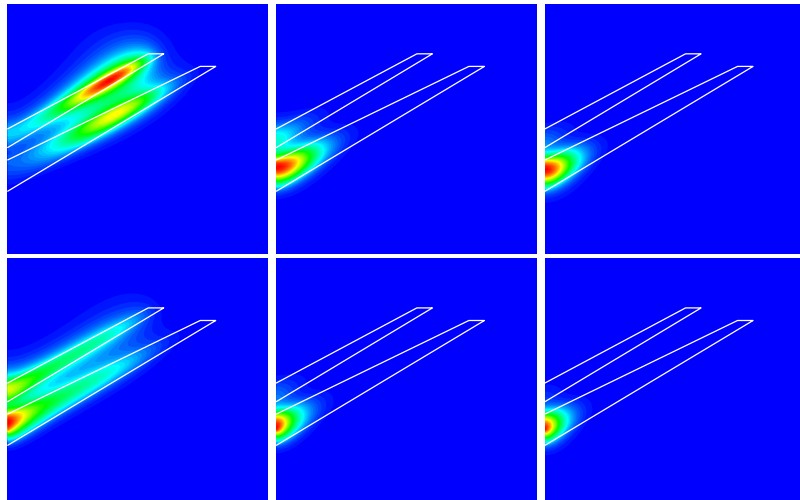

**Figure 3.** The probability density for the lowest confined electron state in a CQDs with $l = 0$. Results are as follows: $F = -75 \, \text{kV/cm}$ (first column), $F = 0$ (second column), and $F = +65 \, \text{kV/cm}$ (third column). The first row is $\kappa = 0$ (without impurity), and the second one with $\kappa = 1$ (with impurity at $z_i = 2.5 \, \text{nm}$). All cases are without magnetic field effects.

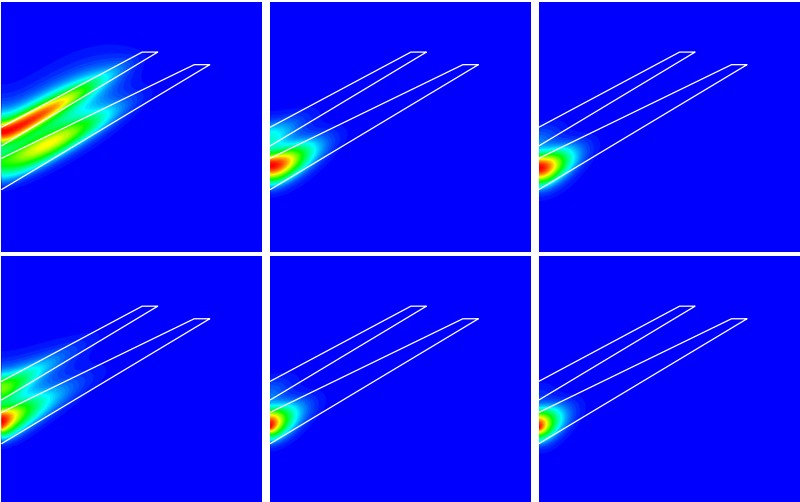

**Figure 4.** The results are as in Figure 3, but for $B = 10 \, \text{T}$.

Figure 2 shows the energy eigenvalues with external electric and magnetic fields, including a shallow donor impurity. Figure 2a,b present the electric field in the range $-80 \, \text{kV/cm}$ to $+80 \, \text{kV/cm}$ without and with impurity, respectively. For the most negative electric field values, the electron moves to the DCQD upper region, see Figure 3 upper left panel, and notice the red region in the upper dot of the structure where the wave function is a maximum. We can interpret this behavior as a deformation of a quantum ring in the upper region of the DCQD. Furthermore, the energy increases with the increasing electric field; consequently, the electron moves steadily towards the DCQD lower region, illustrated in the upper row of Figure 3. As a result, the electron is more confined as the electric field increases. The presence of the impurity is considered in Figures 2b and 3 second row. This causes the energies to decrease almost rigidly, especially for the ground state, due to the additional Coulomb potential. Additionally, the probability density in Figure 3 second row is concentrated on the $z$-axis for $l = 0$ and away from the $z$-axis for $l = \pm 1$ (not shown). Also, when $l = \pm 1$, the energies are degenerate. Figure 2c,d display the magnetic field effects between zero to 30 T without and with impurity, and Figure 4 shows the probability density for $B = 10 \, \text{T}$. In this case, the magnetic field breaks the degeneracy associated with $l = \pm 1$. When $l = 0, +1$, the energies increase with the increasing magnetic field, while

for $l = -1$, the energies start to decrease, and at a certain magnetic field value, they start to increase again; this is related to the third term on the left-hand side of Equation (2). Analogous to Figure 2b, the impurity presence decreases all energies concerning the case without impurity. Comparing the probability density in the upper and lower panels of the first column in Figure 3 with the same panels in Figure 4, it is seen a shift to the left in Figure 4, due to the presence of the magnetic field. In all other cases in Figures 3 and 4 the probability densities are concentrated on the $z$-axis.

Figure 5 shows the combined electric and magnetic field effects for a confined electron in a GaAs DCQD. Figure 5a,b are a reproduction of Figure 2a,b, including only the two first energy states, for comparison. Figure 5b,e display the combined effect of the electric field and $B = 15\,\text{T}$ without and with donor impurity, respectively. One can observe a break of the degeneracy due to the magnetic field presence and the decrease in all state energies due to the impurity Coulomb potential, see Figure 5e. The Figure 5c,f show the same effects but for $B = 30\,\text{T}$. From Figure 5b,e we can see the following characteristics: (i) an increase in the energy values as a function of the applied electric field (the magnetic field produces an additional confinement potential), (ii) the break of the degeneracy for states with $l = \pm 1$, (iii) and that the impurity effect is pulling down the energy values. In Figure 5b,c, it is interesting to note the appearance of a crossing between the ground state and the first excited state. This is evidence of a quantum ring-like behavior under magnetic field effects. The consequence of this crossing is the symmetry change of the ground state wave function. When considering the impurity presence, this crossing disappears, and the system behaves like a quasi-spherical QD under the effects of the applied magnetic field.

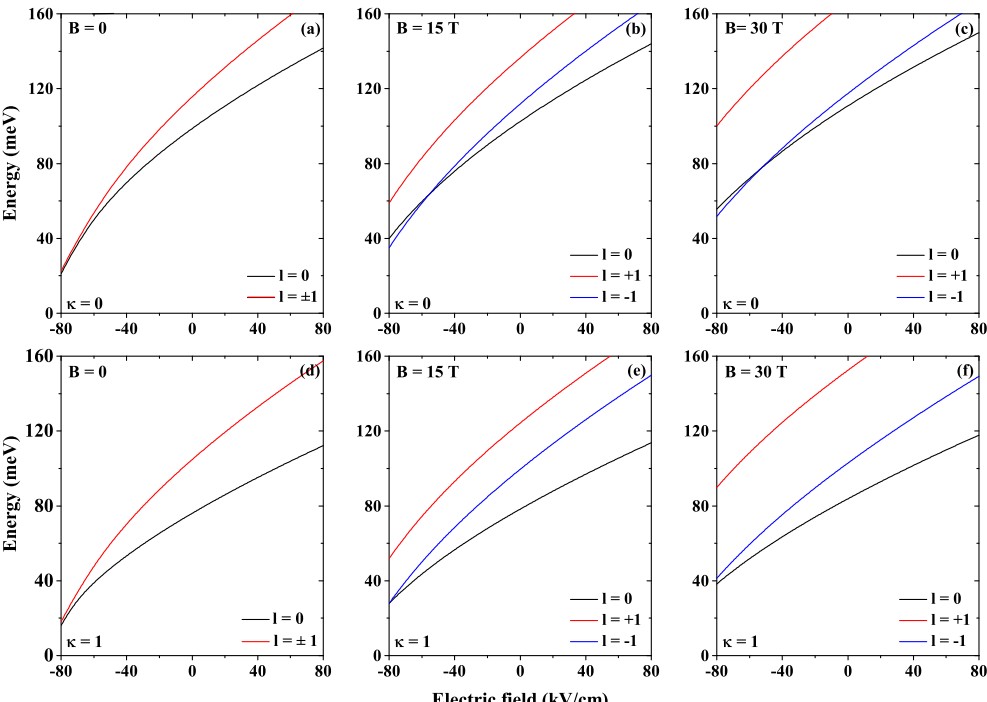

**Figure 5.** The energy of the lowest confined electron states in GaAs DCQD for $l = 0$ and $l = \pm 1$ and under the electric field effects. The results are for three values of the applied magnetic field. The upper row is without impurity effects, whereas in the lower row, a shallow donor is considered at $z_i = 2.5\,\text{nm}$.

Figure 6 depicts the combined effect of a constant electric field and the variation of the magnetic field of the range $B = 0$ to $B = 30\,\text{T}$ for the first three lowest energy states, including the shallow donor impurity effects. Figure 6b,e are a reproduction of Figure 2c,d, for the sake of comparison. Figure 6a,d show the effect of an electric field of $-80\,\text{kV/cm}$. Comparing these figures with Figure 6b,e, the energy values are noticeably lower than

them. On the other hand, when $F = -80\,\text{kV/cm}$, the magnetic field variation increases at a larger rate, producing larger confinement. Figure 6c,f display the effect of an electric field of $+80\,\text{kV/cm}$, which increases the structure's energy values, leading to larger confinement compared to Figure 6a,b,d,e. Also, in Figure 6c,f, the energy increases at a lower rate as a function of the increase in the magnetic field, producing a smaller increase in the relative confinement. The crossing in Figure 6a,d results in a symmetry change in the ground state wave function; this crossing disappears for $F = 0$ and $F = +80\,\text{kV/cm}$. The presence of the impurity in Figure 6d,e,f produces lower energy values due to the extra Coulomb potential.

Given the double-crossing experienced by the states with $l = 0$ and $l = -1$ in Figure 6d, we have decided to show in Figure 7 the corresponding probability density for three values of the applied magnetic field and considering a negative electric field, that is, directed downwards in Figure 1 left. At zero magnetic field, the ground state corresponds to $l = 0$; in this case, the maximum value of the probability density corresponds to the electron located in the upper dot, being located fundamentally in the periphery of the dot but extending towards its central region. In this sense, the structure is of a disk-shaped probability density. A finite probability density in the lower dot allows us to conclude that there is also a high degree of probability in finding the electron in the lower dot. Subsequently, there are two vertically coupled disks. For the case $l = -1$ and zero magnetic fields, the shape of the probability density corresponds to two vertically coupled rings located towards the outside of the two QDs. When applying the magnetic field, it is observed that the probability densities are pushed toward the central region of the two QDs. The state with $l = 0$ happens to have the structure of two vertically coupled quasi-spherical QDs. The size or extent in space decreases as the magnetic field increases, and the probability density at the top dot becomes smaller. This explains why at a field of 20 T the state with $l = 0$ returns to the ground state.

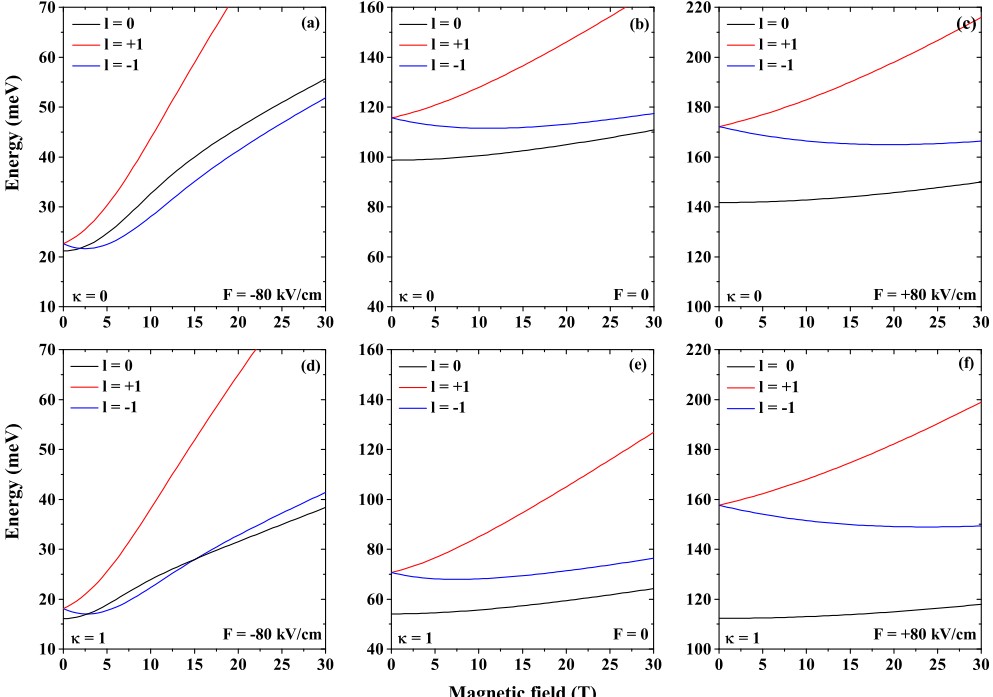

**Figure 6.** The energy of the lowest confined electron states in GaAs DCQD for $l = 0$ and $l = \pm 1$ and under the magnetic field effects. The results are for three values of the applied electric field. The upper row is without impurity effects, whereas in the lower row, a shallow donor is considered at $z_i = 2.5\,\text{nm}$.

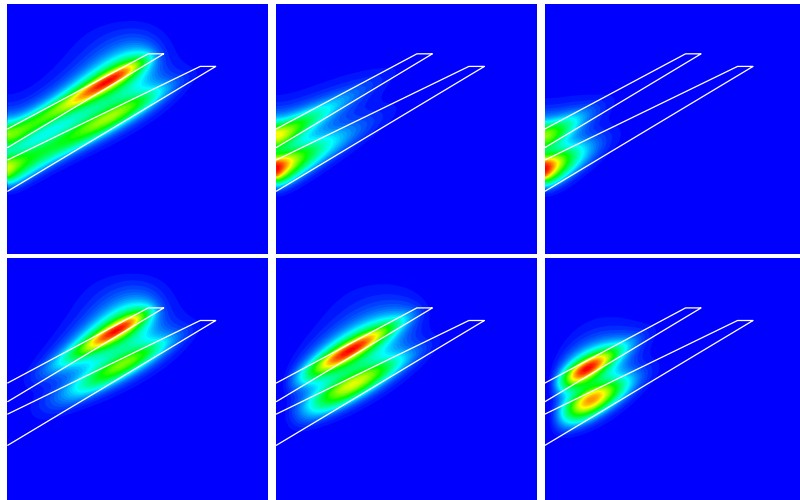

**Figure 7.** The probability density for the lowest confined electron state in a GaAs DCQDs. The upper/lower row is for $l = 0/l = -1$. Results are for $F = -80\,\text{kV/cm}$ and three values of the applied magnetic field: zero (first column), $B = 10\,\text{T}$ (second column), and $B = 20\,\text{T}$ (third column). The impurity is placed at $z_i = 2.5\,\text{nm}$.

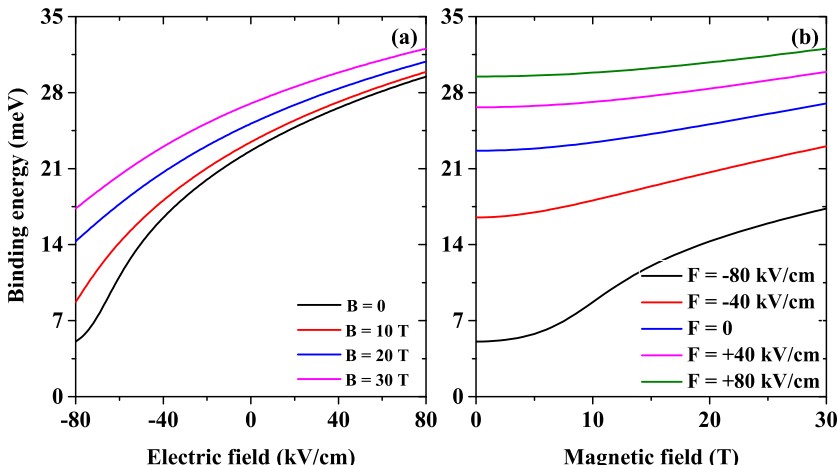

**Figure 8.** The binding energy for the $l = 0$ lowest confined electron state in a CQDs varying the electric/magnetic field with four/five fixed values of the applied magnetic/electric field (**a**,**b**). The impurity is located at $z_i = 2.5\,\text{nm}$.

Figure 8 presents the $l = 0$ ground state binding energy for two cases, varying the electric field for four fixed values of the magnetic field Figure 8a and varying the magnetic field for five fixed values of the electric field Figure 8b. Figure 8a is understood regarding Figure 5, in which, for all cases, the energy increase rate is larger without the impurity being the largest for $B = 0$. For this reason, the binding energy variation is the largest in the latter case. Figure 8b is explained concerning Figure 6 where an analogous energy variation is found as in Figure 8a, but with a lower variation rate, this produces a ground state binding energy that remains constant across the magnetic field range.

Finally, the PCS´s needed parameters and incident photon energy dependencies are presented in Tables 1–3 and in Figures 9 and 10 for $l = 0$ and impurity positions $z_i = 2.5\,\text{nm}$ (at the middle of the lower dot) and $z_i = 6.0\,\text{nm}$ (inside the middle barrier), see the two orange dots in Figure 1 left. The reason for this choice will be evident in the discussion below. Tables 1 and 2 presents the main parameters to calculate the PCS under the effects of an electric/magnetic field. Table 3 presents the main parameters for the combined effects of the magnetic and electric field for the particular case of $z_i = 2.5\,\text{nm}$. Note that the data in the third and fourth columns of Table 1, for the case of $z_i = 2.5\,\text{nm}$, come from the results

in Figure 5a,d, respectively. The data for the fifth and sixth columns are obtained by using Equations (3) and (6), respectively. The data for $z_i = 6.0$ nm are obtained from simulations that we do not report here but are presented to make the article more self-contained. In the case of Table 2, the corresponding information is obtained from Figure 6b,e. Data in Table 3 come from Figure 6a,c.

**Table 1.** Information needed for the calculation of the PCS under electric field effects without the presence of the magnetic field, where $z_i$ is the impurity position, $E_1^0/E_1^1$ is the energy of the ground state without/with the presence of the impurity, and $M_0^1$ is the corresponding matrix element. The calculations are performed with $l = 0$.

| $z_i$ (nm) | $F$ (kV/cm) | $E_1^0$ (meV) | $E_1^1$ (meV) | $E_b$ (meV) | $M_0^1$ (nm) |
|---|---|---|---|---|---|
|  | −80 | 21.1 | 16.1 | 5.0 | 15.7 |
|  | −40 | 69.8 | 53.3 | 16.5 | 6.8 |
| 2.5 | 0 | 98.7 | 76.0 | 22.7 | 5.4 |
|  | 40 | 121.7 | 95.1 | 26.6 | 4.7 |
|  | 80 | 141.7 | 112.2 | 29.5 | 4.2 |
|  | −80 | 21.1 | 15.3 | 5.8 | 16.1 |
|  | −40 | 69.8 | 51.8 | 18.0 | 7.0 |
| 6.0 | 0 | 98.7 | 76.3 | 22.4 | 5.7 |
|  | 40 | 121.7 | 97.3 | 24.4 | 5.0 |
|  | 80 | 141.7 | 116.1 | 25.6 | 4.5 |

**Table 2.** Information needed for the calculation of the PCS under magnetic field effects without the presence of the electric field, where $z_i$ is the impurity position, $E_1^0/E_1^1$ is the energy of the ground state without/with the presence of the impurity, and $M_0^1$ is the corresponding matrix element. The calculations are performed with $l = 0$.

| $z_i$ (nm) | $B$ (T) | $E_1^0$ (meV) | $E_1^1$ (meV) | $E_b$ (meV) | $M_0^1$ (nm) |
|---|---|---|---|---|---|
|  | 0 | 98.7 | 76.0 | 22.7 | 5.4 |
| 2.5 | 10 | 100.5 | 77.1 | 23.4 | 5.3 |
|  | 20 | 104.9 | 79.8 | 25.1 | 5.2 |
|  | 30 | 110.8 | 83.8 | 27.0 | 5.0 |
|  | 0 | 98.7 | 76.3 | 22.4 | 5.7 |
| 6.0 | 10 | 100.5 | 77.4 | 23.1 | 5.6 |
|  | 20 | 104.9 | 80.4 | 24.5 | 5.5 |
|  | 30 | 110.8 | 84.7 | 26.1 | 5.3 |

**Table 3.** Information needed for the calculation of the PCS under combined electric and magnetic field effects and considering the presence of a donor impurity at $z_i = 2.5$ nm. Here $E_1^0/E_1^1$ is the energy of the ground state without/with the presence of the impurity, and $M_0^1$ is the corresponding matrix element. The calculations are performed with $l = 0$.

| $B$ (T) | $F$ (kV/cm) | $E_1^0$ (meV) | $E_1^1$ (meV) | $E_b$ (meV) | $M_0^1$ (nm) |
|---|---|---|---|---|---|
| 10 | −80 | 32.6 | 23.9 | 8.7 | 10.5 |
| 30 | −80 | 55.6 | 38.3 | 17.3 | 7.8 |

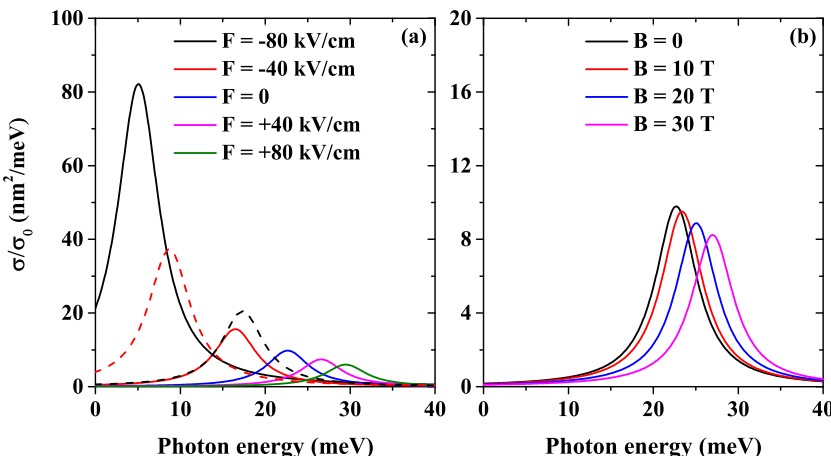

**Figure 9.** Shallow donor impurity related PCS as a function of the incident photon energy for a confined electron in a GaAs DCQDs. In (**a**), the results are as follows: solid lines are for several values of the applied electric field with $B = 0$, the dashed-red line is for $F = -80\,\text{kV/cm}$ with $B = 10\,\text{T}$, and the dashed-black line corresponds to $F = -80\,\text{kV/cm}$ with $B = 30\,\text{T}$. In (**b**), the results are for several values of the applied magnetic field with $F = 0$. The impurity is located at $z_i = 2.5\,\text{nm}$, and transitions only consider the impurity ground state with $l = 0$.

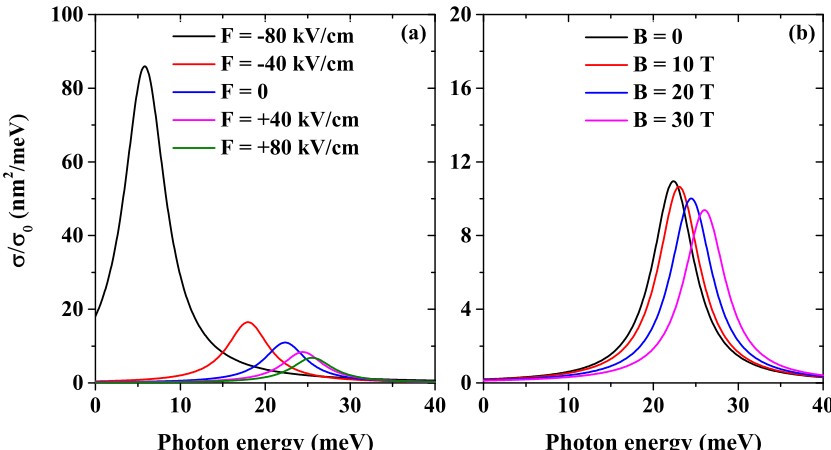

**Figure 10.** Shallow donor impurity related PCS as a function of the incident photon energy for a confined electron in a DCQDs. In (**a**), the results are for several values of the applied electric field with $B = 0$, whereas in (**b**), they are for several values of the applied magnetic field with $F = 0$. The impurity is located at $z_i = 6.0\,\text{nm}$, and transitions only consider the impurity ground state with $l = 0$.

In Figure 9 the way in impurity-related PCS varies is analyzed as a function of the incident photon energy for different electric field values and a mixture of electric-magnetic fields, Figure 9a, and different magnetic field values, Figure 9b. In both cases, the impurity is placed at $z_i = 2.5\,\text{nm}$. For the electric field magnitudes taken in Figure 9a ($F = -80\,\text{kV/cm}$, $F = -40\,\text{kV/cm}$, $F = 0$, $F = +40\,\text{kV/cm}$, and $F = +80\,\text{kV/cm}$), it is seen that the PCS is the largest for the more negative field and decreases by almost a factor of eight for the largest and positive electric field, where the reduction of optical integral magnitude plays an important role (see Table 1). From Equation (5), it is easy to notice that the maximum cross section is reached when the binding energy is equal to the incident photon energy. Table 1 shows the increase of the binding energy maximum. Figure 9b shows the PCS as a function of the incident photon energy for several values of the applied magnetic field ($B = 0$, $B = 10\,\text{T}$, $B = 20\,\text{T}$, and $B = 30\,\text{T}$). In this case, the maximum peak slightly decreases as the magnetic field increases (see Table 2); the optical integral gives similar magnitudes. Additionally, Figure 9a displays the combined effect of electric and magnetic fields, dotted lines, on the PCS for two cases. A fixed electric field of magnitude $F = -80\,\text{kV/cm}$ and two

values of the magnetic field, $B = 10\,\text{T}$ and $B = 30\,\text{T}$. For $B = 10$ the PCS magnitude is half the largest PCS with only an electric field. For $B = 30\,\text{T}$ the PCS magnitude is half that for $B = 10\,\text{T}$.

In this work, $\Gamma = 3.0\,\text{meV}$ has been used as a constant value throughout the manuscript, regardless of how close or far the transition energy from the ground impurity state to the ground electron state is to zero. Of course, the closer that energy difference is to zero, a much smaller $\Gamma$-parameter value should be considered to avoid a possible misunderstanding by showing the photoionization curves presenting a non-zero value when the photon energy is zero (see the solid-black and dashed-red curves in Figure 9a).

Figure 10 displays the PCS as a function of the incident photon energy for different electric field values, Figure 10a, and different magnetic field values, Figure 10b. In both cases, the impurity is placed at $z_i = 6.0\,\text{nm}$. For the electric field magnitudes taken in Figure 8a, it is seen that the PCS is the largest for the more negative electric field and decreases strongly for the other cases; the trend for the optical integral magnitude is close to the case in Figure 9, similar to the case with the impurity at $z_i = 2.5\,\text{nm}$. Table 1 shows the increase of the binding energy maximum. Figure 10b shows the PCS as a function of the magnetic field for the same values taken in Figure 9; in this case, the maximum peak is quite constant for all values (see Table 2), with a small decrease as the magnetic field increases, and also, the PCS magnitude is smaller by a factor of ten than for the electric field.

## 4. Conclusions

In this study, we examined the effects of electric and magnetic fields and a donor impurity, at different positions, on the photoionization cross-section of a vertical conical double quantum dot heterostructure. The effective mass, parabolic conduction band approximations, and FEM were used to obtain the electronic structure.

An electric field was applied from $-80\,\text{kV/cm}$ to $+80\,\text{kV/cm}$ modifying the electronic confinement. Depending on the sign of the electric field, the impact on probability density was shown, evolving from an annular character to a punctiform one. The presence of the applied magnetic field breaks the degeneracy between $l = \pm 1$. In this case, the probability density is pulled towards the axial axis, indicating greater electron confinement. The effect of the donor impurity presence further increases this magnetic field confinement. When the impurity was present in the system, decreased energy values due to the Coulomb potential could be observed. Concerning the PCS, the simulations show that the binding energy is the dominant factor; the resonant peaks correspond to the incident photon energy equal to the donor impurity binding energy. Depending on the electric and magnetic field values and the impurity position, they are blueshifted or redshifted. The results of the two impurity positions are very similar, with small binding energy dispersion and small changes in the peak magnitude. The combined effects of electric and magnetic fields were illustrated for a fixed electric field with three values for the magnetic field, producing a very similar behavior on the PCS as for the electric field alone case.

Vertically coupled quantum dots are very promising for designing many devices due to flexibility in selecting a geometry, choosing different individual dots, and changing the coupling between them, which leads to applications including single photon sources, entangled quantum pairs, qubits, and gates in quantum computation, quantum cryptography, solar cells, lasers, LEDS, biomedical imaging, and drug delivery systems.

**Author Contributions:** A.M.L.A., F.M.R.: conceptualization, methodology, software, formal analysis, investigation, supervision, writing; Á.L.M., J.A.V.: methodology, software, formal analysis; C.A.D.: formal analysis, writing. All authors have read and agreed to the published version of the manuscript.

**Funding:** The authors are grateful to Colombian agencies CODI-Universidad de Antioquia (Estrategia de Sostenibilidad de la Universidad de Antioquia and projects "Propiedades magneto-ópticas y óptica no lineal en superredes de Grafeno", "Estudio de propiedades ópticas en sistemas semiconductores de dimensiones nanoscópicas", "Propiedades de transporte, espintrónicas y térmicas en el sistema molecular ZincPorfirina", and "Complejos excitónicos y propiedades de transporte en sistemas

nanométricos de semiconductores con simetría axial") and Facultad de Ciencias Exactas y Naturales-Universidad de Antioquia (ALM and CAD exclusive dedication projects 2022–2023).

**Institutional Review Board Statement:** Not applicable.

**Informed Consent Statement:** Not applicable.

**Data Availability Statement:** No new data were created or analyzed in this study. Data sharing is not applicable to this article.

**Conflicts of Interest:** The authors declare no conflict of interest.

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
