# Peer review of "Electric and Magnetic Fields Effects in Vertically Coupled GaAs/AlxGa1−xAs Conical Quantum Dots"

_condensedmatter, doi:10.3390/condmat8030071_

Round 1

Reviewer 1 Report

The theoretical paper is devoted to the study of the energy electronic structure in double coupled conical GaAs/AlGaAs quantum dots in the presence of a Coulomb  center in the structure, which does not violate the axial symmetry of the system. The effect of external electric and magnetic fields on the level structure and photoionization spectra is studied. In the work, new scientific results were obtained, which were of interest from the point of view of the development of the physics of zero-dimensional systems. The article may be published subject to the following remarks.

1. When setting the research problem, it is desirable to substantiate the possibility of manufacturing such coupled quantum dots of rather complex geometry. How realistic is it in practice to control the position of the Coulomb center in a quantum dot?

2. How realistic is it to obtain a uniform electric field in the complex system under study, which includes space charge regions?

3. For what temperature are the calculations made? The different filling of the initial and final states at nonzero temperature can affect the photoionization spectra.

4. The value of the broadening parameter is 1 meV; however, the spectra in Figs. 9-11 are more broadened. What is the reason?

5. How correct is it to show in Fig. 9-11 quantum energy values equal to 0?

6. You should check the correctness of the reference [21] before the formula (4).

7. It is desirable to clarify that formula (4) uses the effective and average fields of the light wave, and not the applied constant field.

8. Line 55. Replace "Colombian" with Coulombian" :-)

Author Response

Referee 1

The Referee:

The theoretical paper is devoted to the study of the energy electronic structure in double coupled conical GaAs/AlGaAs quantum dots in the presence of a Coulomb center in the structure, which does not violate the axial symmetry of the system. The effect of external electric and magnetic fields on the level structure and photoionization spectra is studied. In the work, new scientific results were obtained, which were of interest from the point of view of the development of the physics of zero-dimensional systems. The article may be published subject to the following remarks.

Our reply:

We want to thank the Referee for his/her observations and comments, which have helped us to substantially improve the quality of our manuscript. We hope that our responses and changes in the revised version of the manuscript are satisfactory and that our article is suitable for publication in the Condensed Matter Journal.

The Referee:

  1. When setting the research problem, it is desirable to substantiate the possibility of manufacturing such coupled quantum dots of rather complex geometry. How realistic is it in practice to control the position of the Coulomb center in a quantum dot?

Our reply:

We thank the Referee for his/her comment. We want to emphasize that the type of vertically coupled systems that we present in this study can be implemented experimentally using the Local Droplet Etching technique. In the last paragraph of the Introduction section we have included the following text that also includes a comment concerning the intentional doping (the corresponding references have been included):

The system of two vertically coupled conical quantum dots has been reported experimentally by Heyn and coworkers \cite{33,34}. Using the local droplet etching (LDE) technique, where strain-free and widely adjustable GaAs quantum-dot molecules (QDMs) can be synthesized, they studied the excited-state indirect excitons in GaAs quantum dot molecules. Regarding the impurities located along the axial axis considered in this study, it is important to clarify that this is one of the most particular cases of the problem to be implemented. Although it is possible to establish an approximate region where impurities can be located within the structure, intentional doping is still a technique in development. A more extensive theoretical study should consider random doping with impurities within the structure, including acceptor impurities.

The Referee:

  1. How realistic is it to obtain a uniform electric field in the complex system under study, which includes space charge regions?

Our reply:

We thank the Referee for his/her comment. The study of the effects of electric fields in this type of system has already been reported experimentally by the group of Heyn et al. In Ref. [34] of the revised version of this manuscript the authors present a study of electric field effects in conical quantum dots where the transition from a ring system to a quantum dot system can be induced. The implementation of tilted electric field effects and axially applied magnetic fields is in the process of development by the same group of collaborators.

The Referee:

  1. For what temperature are the calculations made? The different filling of the initial and final states at nonzero temperature can affect the photoionization spectra.

Our reply:

In the first paragraph of the Results and Discussion section we added the following comment:

Calculations are at $T=4$\,K. For finite temperature values, the different filling of the initial and final states can affect the photoionization cross-section.

The Referee:

  1. The value of the broadening parameter is 1 meV; however, the spectra in Figs. 9-11 are more broadened. What is the reason?

Our reply:

We thank the Referee for his/her observation. We are sorry for a possible misunderstanding. The correct Gamma-parameter value is 3.0 meV (see the second line of the first paragraph of the Results and Discussion section of the revised version of the manuscript.) This Gamma-parameter value is consistent with the broadening of the curves in Figs 9-11.

The Referee:

  1. How correct is it to show in Fig. 9-11 quantum energy values equal to 0?

Our reply:

After the discussion of Fig. 9, we have added the following comment:

In this work, $\Gamma=3.0$\,meV has been used as a constant value throughout the manuscript, regardless of how close or not the transition energy from the ground impurity state to the ground electron state is to zero. Of course, the closer that energy difference is to zero, a much smaller $\Gamma$-parameter value should be considered to avoid a possible misunderstanding by showing the photoionization curves presenting a non-zero value when the photon energy is zero (see the solid-black and dashed-red curves in Fig. 9(a)).

The Referee:

  1. You should check the correctness of the reference [21] before the formula (4).

Our reply:

We thank the Referee for this observation. We have made the appropriate correction and the relevant references to Eq. (4) are cited in the revised version of the manuscript.

The Referee:

  1. It is desirable to clarify that formula (4) uses the effective and average fields of the light wave, and not the applied constant field.

Our reply:

We thank the Referee for his suggestion. After the Eq. (4), we have added the following comment:

“We note that in Eq. (4) is used the effective and average fields of the light wave and not the externally applied constant electric field.”

The Referee:

  1. Line 55. Replace "Colombian" with Coulombian" :-)

Our reply:

The typo has been amended.

Reviewer 2 Report

Line 40- Check the symbol of element: CadSe/ZnS quantum dots,

Line 111- How do you decide x = 0.3

- There are no citation of any reference for equations 1 to 6. Cite some references for these mathematical equations

-Line 157: What is the basis on which the parameters are choosen.

- Table 1, 2 and 3: From where do you find the value presented in the tables and is there any figure that are related to these tables.

-Conclusions: Its too lengthy and need to be reduced.

Moderate editing of English language required

Author Response

Referee 2

The Referee:

Line 40- Check the symbol of element: CadSe/ZnS quantum dots,

Our reply:

The typo has been amended.

The Referee:

Line 111- How do you decide x = 0.3

Our reply:

We thank the Referee for his/her question. In the revised version of the manuscript we have included two experimental references, [33,34], where the authors report the growth of heterostructures with shape, size, and stoichiometric characteristics similar to those we have modeled in this article. So, the motivation for x=0.3 remains in experimental works.

The Referee:

- There are no citation of any reference for equations 1 to 6. Cite some references for these mathematical equations

Our reply:

We thank the Referee for his/her suggestion. Some References have been included for the main equations.

The Referee:

-Line 157: What is the basis on which the parameters are choosen.

Our reply:

In the first paragraph of the Results and Discussion section we have included the four References from which we have taken the parameters to simulate our calculations. References in the revised version of the manuscript are [19,26,27,35].

The Referee:

- Table 1, 2 and 3: From where do you find the value presented in the tables and is there any figure that are related to these tables.

Our reply:

In the paragraph that follows the discussion of Fig. 8, we have added the following comment:

Note that the data in the third and fourth columns of Table 1, for the case of $z_i=2.5$\,nm, come from the results in Figs. 5(a) and 5(d), respectively. The data for the fifth and sixth columns is obtained by using Eqs. (3) and (6), respectively. The data for $z_i=6.0$\,nm are obtained from simulations that we do not report here but are presented to make the article more self-contained. In the case of Table 2, the corresponding information is obtained from Figs. 6(b) and 6(e). Data in Table 3 come from Figs. 6(a) and 6(c).

The Referee:

-Conclusions: Its too lengthy and need to be reduced.

Our reply:

The Conclusions section was substantially modified to reduce its length and present the findings in a more general way.

We want to thank the Referee for his/her observations and comments, which have helped us to substantially improve the quality of our manuscript. We hope that our responses and changes in the revised version of the manuscript are satisfactory and that our article is suitable for publication in the Condensed Matter Journal.

Round 2

Reviewer 2 Report

Accept in present form